# Phenotypic Characterization of Recombinant Marek’s Disease Virus in Live Birds Validates Polymorphisms Associated with Virulence

**DOI:** 10.3390/v15112263

**Published:** 2023-11-16

**Authors:** Taejoong Kim, Cari J. Hearn, Jody Mays, Deborah Velez-Irizarry, Sanjay M. Reddy, Stephen J. Spatz, Hans H. Cheng, John R. Dunn

**Affiliations:** 1Southeast Poultry Research Laboratory, U.S. National Poultry Research Center, Agricultural Research Service, U.S. Department of Agriculture, 934 College Station Road, Athens, GA 30605, USA; stephen.spatz@usda.gov (S.J.S.); john.dunn@usda.gov (J.R.D.); 2Avian Diseases and Oncology Laboratory, U.S. National Poultry Research Center, Agricultural Research Service, U.S. Department of Agriculture, 4279 E. Mount Hope Road, East Lansing, MI 48823, USA; cari.hearn@usda.gov (C.J.H.); jody.mays@usda.gov (J.M.); deb.velezirizarry@usda.gov (D.V.-I.); hans.cheng@usda.gov (H.H.C.); 3Department of Veterinary Pathobiology, Texas A&M University, College Station, TX 77843, USA; sreddy@tamu.edu

**Keywords:** Marek’s disease, single-nucleotide polymorphisms, virulence, viral genome

## Abstract

Marek’s disease (MD) is a highly infectious lymphoproliferative disease in chickens with a significant economic impact. *Mardivirus gallidalpha* 2, also known as Marek’s disease virus (MDV), is the causative pathogen and has been categorized based on its virulence rank into four pathotypes: mild (m), virulent (v), very virulent (vv), and very virulent plus (vv+). A prior comparative genomics study suggested that several single-nucleotide polymorphisms (SNPs) and genes in the MDV genome are associated with virulence, including nonsynonymous (ns) SNPs in eight open reading frames (ORF): UL22, UL36, UL37, UL41, UL43, R-LORF8, R-LORF7, and ICP4. To validate the contribution of these nsSNPs to virulence, the vv+MDV strain 686 genome was modified by replacing nucleotides with those observed in the vMDV strains. Pathogenicity studies indicated that these substitutions reduced the MD incidence and increased the survival of challenged birds. Furthermore, using the best-fit pathotyping method to rank the virulence, the modified vv+MDV 686 viruses resulted in a pathotype similar to the vvMDV Md5 strain. Thus, these results support our hypothesis that SNPs in one or more of these ORFs are associated with virulence but, as a group, are not sufficient to result in a vMDV pathotype, suggesting that there are additional variants in the MDV genome associated with virulence, which is not surprising given this complex phenotype and our previous finding of additional variants and SNPs associated with virulence.

## 1. Introduction

Marek’s disease virus (MDV), *Mardivirus gallidalpha* 2, is the causative pathogen of Marek’s disease (MD), which is ubiquitous in the environment and also highly infectious to chickens. MD is characterized by T-cell tumors, lymphoid organ atrophy, the enlargement of peripheral nerves, and the clinical symptoms of transient paralysis, mortality, and immunosuppression in susceptible birds [1,2,3,4]. MDV has repeatedly evolved to be more virulent since the introduction of MD vaccines in the 1970s, which can destroy the efficacy of existing MD vaccines in 10–20-year cycles [5,6]. While effective in preventing disease, MD vaccines do not prevent viral replication and allow re-infection and replication of the field strains in vaccinated birds, which likely promotes the creation of new MDV strains with improved lifetime transmission potential [7]. Thus, imperfect MD vaccines are widely speculated to be the main driving force in this viral evolution [7,8,9]. The virulence of MDV strains is categorized based on the pathotype from non-virulent to highly virulent MDV [1,5,10,11]. The pathotyping assay measures the ability of a candidate MDV strain to cause MD in vaccinated chickens given defined MD vaccines and compares it with prototype MDV strains that are virulent (v), very virulent (vv), and very virulent plus (vv+) pathotypes [5]. 

A major goal of the field is to identify polymorphisms in the MDV genome that are associated with virulence, which may aid further efforts to design more effective MD vaccines, identify potentially new and more virulent MDV strains without the need for pathotyping, and determine genes and amino acids associated with immune escape. Single-nucleotide polymorphisms (SNPs) were previously identified in the MDV genome using targeted or full-genome sequencing analysis from various isolates [12]. Over many years, 70 MDV strains from various regions in the US have been characterized for virulence, including 19, 24, and 27 strains of vMDV, vvMDV, and vv+MDV pathotypes, respectively. Following analyses, 24 SNPs were associated with virulence, which included 9 nonsynonymous (ns) SNPs in eight ORFs: UL22 (glycoprotein H), UL36 (large tegument protein), UL37 (tegument protein), UL41 (virion shot shutoff protein), UL43 (probable membrane protein), R-LORF8 (nuclear localization signal), R-LORF7 (Meq protein, transcription factor), and ICP4 (transcription factor). In addition, an additional three ORFs, UL6 (capsid protein), UL15 (DNA packaging protein), and SORF1 (unknown function), were also found to be associated with virulence [12]. 

To validate whether these nsSNPs are genuinely associated with virulence, we modified the vv+MDV 686 bacterial artificial chromosome (BAC) clone (also known as 686-BAC) by two-step Red-mediated recombineering to produce recombinant MDVs, v686 mut1 and v686 mut2, which replaced all nine nsSNPs in the allele associated with the majority of vMDV strains. When birds were challenged with these viruses, the virulence of both viruses was found to be reduced compared to the parental 686-BAC-derived virus. To determine virulence more precisely, both viruses were evaluated using a modified best-fit pathotyping assay, which determined them to be vvMDV pathotypes. Thus, this strongly supports that one or more of the nine nsSNPs in the eight MDV genes are associated with virulence. However, it also shows that, as a group, they are insufficient to achieve a vMDV pathotype, indicating that other SNPs, especially those previously identified, also probably contribute to virulence, which is unsurprising given the complex phenotype. Still, this study proves the need for and power of validating genomic screens such as those that identify candidate polymorphisms associated with virulence in the MDV genome. 

## 2. Materials and Methods

### 2.1. Viruses

MD vaccines, the turkey herpesvirus (HVT) Fc126 strain and MDV serotype 2 vaccine SB-1 strain, and MDV prototypes, JM/102W strain (v pathotype), Md5 strain (vv pathotype), and 648A strain (vv+ pathotype), were obtained from the repository of the Avian Disease Oncology Laboratory (ADOL), USDA-ARS (East Lansing, MI, USA) and these viruses were used as vaccine or reference strains for the pathotyping assay [13,14]. Chicken embryo fibroblasts (CEFs) from ADOL line 0 embryos were used to propagate recombinant MDVs [15,16].

### 2.2. Mutagenesis of 686-BAC

The substitution of 9 SNPs into the 8 ORFs (Table 1) was conducted sequentially in a 686-BAC clone [17] with two-step Red-mediated recombination [18]. Since three of the ORFs, R-LORF8, R-LORF7, and ICP4, were found within repeat regions of the MDV genome, an approximately 17 kb region of 686-BAC spanning the terminal repeat short (TRS) and terminal repeat long (TRL) from ICP4 to R-LORF8 was first deleted to generate a clone having only one copy of each of these three ORFs using FLP recombinase-mediated gene deletion [19]. The primers containing the homologous flanking arms necessary for recombineering are listed in Appendix A. Following the deletion of the diploid 3 ORFs, the 686-BAC clone with deleted TRS-TRL (686∆TRS-TRL BAC) was modified at the 9 SNPs in the 8 ORFs with donor fragments, which were amplified PCR amplicons generated using the pEP-KanS template with specific primers (Table 2). Finally, the SNP-modified BAC clones with two-step recombineering were analyzed via digestion with a restriction endonuclease. Approximately 1 μg of BAC clone DNA prepared using the NucleoBond Xtra BAC DNA purification kit (Macherey-Nagel, Duren, Germany) was digested with EcoRV (New England Biolabs., Ipswich, MA, USA) at 37 °C for 3 h and separated on 1× TAE agarose gels. The digested fragments of 686-BAC DNA were stained with Gel Red (Phenix Research, NJ, USA) and visualized using the ChemiDoc XRS gel imager (BioRad Lab., Hercules, CA, USA).

The modified BAC clones were analyzed to validate the SNP changes by next-generation sequencing. Briefly, DNA from 4 BAC samples (686, 686∆TRS-TRL, 686 mut1, and 686 mut2) was submitted to the Michigan State University Research Technology Support Facility (RTSF) for next-generation sequencing. Libraries were prepared using the Roche Kapa HyperPrep DNA Library Preparation Kit with Kapa Unique Dual Index adapters following the manufacturer’s recommendations. Completed libraries were checked for quality and quantity using a combination of Qubit dsDNA HS and Agilent 4200 TapeStation HS DNA1000 assays. The four libraries were combined in equimolar amounts, and the pool was quantified using the Invitrogen Collibri Quantification qPCR kit. The pool was loaded onto a MiSeq v2 Nano flow cell, and sequencing was performed in a 2 × 150 bp paired-end format using a MiSeq v2 300 cycle reagent cartridge. Base calling was performed by Illumina Real Time Analysis (RTA) v1.18.54 and the output of RTA was demultiplexed and converted to FastQ format with Illumina Bcl2fastq v2.20.0. 

Illumina adapter sequences and low-quality bases were removed using Trimmomatic [20], discarding the first six leading bases and bases with Phred quality scores below 15. Filtered reads were mapped to the reference Md5 synthetic contract (RefSeq HQ149526.1) using HISAT2 [21] with default parameters and suppressed discordant alignments for paired reads. SNP calls were generated using BCFtools mpileup [22] with base quality scores above 25. 

### 2.3. Reconstitution of the 686 Mut Viruses

As described previously, the 686∆TRS-TRL BAC and SNP-modified 686-BAC viruses were reconstituted by transfecting BAC DNA into CEFs [16]. Six days post-transfection, virus recovery was confirmed by virus plaque formation in CEFs. The reconstituted viruses were further propagated in CEFs and stored at −140 °C for future use. Passage 7 of 3 recombinant viruses (686-BAC virus with a single deleted copy of the terminal repeats (v686∆TRS-TRL) or 686-BAC viruses with modified single nucleotides in the 8 ORFs (v686 mut1 and v686 mut2)) was used. The nucleotide modifications in recombinant MDVs were verified by Sanger sequencing of the PCR products from virus-infected CEFs. The target regions containing the SNP-modified ORFs were amplified with primer sets (Appendix A) using *pfx* DNA polymerase (Invitrogen, Carlsbad, CA, USA) and Sanger sequenced to confirm the substitution of the target nucleotides. The restoration of the deleted TRS and TRL region was confirmed by PCR amplification with the primer set ICP4.FP 5′-GCCCCTCCTAAACC-CTAA-3 and TRL.RP 5′-GGTCGTCTACTGTTTGTGG-3′ using *pfx* DNA polymerase. 

### 2.4. In Vivo Pathogenicity Study

Animal studies were conducted to measure the virulence of the SNP-modified viruses relative to the parental virus. All experiments were approved by the Institutional Animal Care and Use Committee of the ADOL, US National Poultry Research Center (USNPRC), USDA-ARS. The relative MDV virulence was evaluated in two trials that used the same bird genetics but differed as to whether the chicks had maternal antibodies to MD vaccines or not. In the first trial, 75 maternal antibody-negative ADOL 15I_5_ × 7_1_ birds were randomized into five groups of 15 chickens each and housed in Horsfall–Bauer isolators with wing bands. At five days of age, chickens were inoculated with 500 plaque-forming units (pfu) by the intraabdominal route with one of the following: 686-BAC virus (v686-BAC), 686-BAC virus with a single deleted copy of the terminal repeats (v686∆TRS-TRL) or 686-BAC viruses with modified single nucleotides in the 8 ORFs (v686 mut1 and v686 mut2). Age-matched uninfected chickens as a control group were housed under the same conditions. Birds were monitored daily for 8 weeks post-challenge in all experimental groups. At necropsy or when removed due to mortality or having reached the humane endpoint, MD was scored by evaluating MD macroscopic lesions, e.g., atrophy of the thymus (TA) and bursa of Fabricius (BA), the enlargement of the vagus, brachial, or sciatic nerves (lesion score from 1 to 4; 4 is most severe), and/or tumor formation in internal organs. The second trial was identical to the first except that the chickens were ADOL 15I_5_ × 7_1_ maternal-antibody-positive for MD vaccines.

### 2.5. Pathotyping Assay

The pathotype of the SNP-modified viruses was compared to prototype MDVs with the modified best-fit assay [13,14]. Two replicates were conducted to ensure the reproducibility of the results. All experiments were approved by the Institutional Animal Care and Use Committee of the USNPRC, USDA-ARS. Chickens from a commercial white leghorn SPF flock (Line 22, Charles River Laboratories International, Inc. Wilmington, MA, USA) were randomly divided at hatching into 18 groups, with 3 groups per challenge virus, and housed in Horsfall–Bauer isolators. All birds were wing-banded and had access to feed and water ad libitum. Each virus being evaluated for virulence required the intraabdominal inoculation of birds with different MD vaccines at one day of age: non-vaccinated, 2000 pfu HVT-vaccinated (Fc126 strain), or 1000 pfu HVT and 1000 pfu SB1 (serotype 2) bivalent-vaccinated. At five days of age, birds were intraabdominally challenged with 500 pfu of JM/102W, Md5, 648A, v686∆TRS-TRL (parental strain control), v686 mut1, or v686 mut2. All birds were monitored daily for 8 weeks post-challenge, and MD clinical signs and disease incidence were evaluated as described above.

### 2.6. Determination of Pathotype

The weighted mean of the total MD percentage from JM/102W, Md5, 648A, v686∆TRS-TRL, v686 mut1, and v686 mut2 was calculated from the total MD percentage in HVT-vaccinated and bivalent-vaccinated groups, with the bivalent-vaccinated MD% calculated with two-fold weight. The formula to calculate the weighted mean of the challenge viruses was as follows: Weighted Mean = (% MD in HVT vaccinated + % MD in bivalent vaccinated + % MD in bivalent vaccinated)/3

The calculated weighted mean was used to rank the virulence of v686 mut1 and v686 mut2, and the virulence distance of MDV was calculated proportionally between JM/102W and Md5 using the mean of the percentage total MD in non-vaccinated chickens or between Md5 and 648A using the weighted mean [13]. The formula to calculate the proportional distance of 686 viruses was as follows: Proportional distance = (% response of the isolate − % response of the less virulent reference strain)/(% response of the more virulent strain − % response of the less virulence strain)

The SNP-modified viruses were categorized based on the comparison with the reference prototype strains. The pathotype was assigned as the reference strain’s pathotype that was most similar with respect to critical parameters and MD incidences [14].

### 2.7. Statistical Analysis

Statistical analysis was performed with the GraphPad Prism (ver. 10, GraphPad Software, Boston, MA, USA) package. A comparison of survival curves in the vaccine-challenge groups was performed with Log-rank (Mantel–Cox) tests. Two-way ANOVA analysis with Tukey’s multiple-comparison test was used to compare the MD incidence among the experimental groups. The statistically significant results were based on a *p*-value < 0.05.

## 3. Results

### 3.1. Modified Recombinant MDVs

The 686-BAC used in this study is an infectious BAC clone containing the entire genome of the vv+MDV strain 686. To alleviate the need to modify both copies of R-LORF8, R-LORF7, and ICP4, 686∆TRS-TRL BAC was constructed by introducing a deletion into one copy of the TRS/TRL subgenomic region from ICP4 to R-LORF8, which was confirmed by restriction digestion profiling: e.g., a 20.5 kb fragment (686-BAC) was reduced to a 7.8 kb fragment in the 686∆TRS-TRL clones (Figure 1, Lane 1 and Lane 2). Notably, there were no restriction length differences in the EcoRV patterns of two SNP-modified 686-BAC clones (686 mut1 and 686 mut2) that were generated in independent experiments using the 686∆TRS-TRL BAC (Figure 1, Lanes 2–4). Next-generation sequence files consisted of ~149 k short reads per library and a mean Phred quality score of 36. On average, 63% of quality-filtered sequence reads mapped to the reference genome at an average base coverage of 145. SNPs called with a minimum read depth of 10 or less and strand bias across all samples were excluded from further analysis. The retained SNPs included the nine nsSNPs (Appendix A) and two additional SNPs with an average read depth of 138 (Table 3). One of the additional SNPs identified mapped six bases upstream of the mutated UL43 locus. The G-to-T base substitution altered the amino acid at position 397 of the UL43 protein from alanine to serine. This additional SNP was found only in the mutated BAC clones with the amino acid sequence altered at positions 397–399 to SerTyrVal from AlaTyrIle in the 686 and 686∆TRS-TRL BACs (Appendix A). The second additional SNP was found only in the 686∆TRS-TRL BAC and mapped 75 bases upstream of LORF2-coding sequences. The specificity of the upstream LORF2 SNP to only the 686∆TRS-TRL BAC and not any of the mutants may point to a sequencing error rather than a true SNP. Sequences were deposited as GenBank accession numbers OR813908, OR813909, OR813906, and OR813907 for 686∆TRS-TRL BAC, for 686 mut1, and for 686 mut2, respectively.

A total of three recombinant 686-BAC viruses, a 686-BAC virus with a TRS-TRL deletion (v686∆TRS-TRL) and two SNP-modified viruses containing all SNP modifications (v686 mut1 and v686 mut2), were recovered and propagated in fresh CEFs. v686 mut1 and v686 mut2 were generated independently by introducing nucleotide substitution consecutively from UL22 to ICP4, as shown in Table 1. The nucleotide sequences of PCR products from v686 mut1 and v686 mut2 indicated that all manipulations were successful in generating SNP-modified 686-BAC viruses, as described in Table 2 (Appendix A). The deleted TRS and TRL region of recombinant 686 viruses was restored by passage in CEFs (Appendix A).

### 3.2. Pathogenicity Analysis of the Recombinant MDVs

The 686-BAC-derived viruses showed a high degree of virulence in both trials. The deletion of ICP4 to R-LORF8 in the MDV strain 686 genome did not affect the mortality of the v686∆TRS-TRL-infected group. In trial 1 with antibody-negative birds, the 50% mortality of v686∆TRS-TRL occurred at 34 days post-infection (dpi), which was similar to that of v686-BAC (Figure 2). At the same timepoint, the survival rates of the v686 mut1 group and the v686 mut2 group were higher. As expected, the survival rates of v686 mut1 and v686 mut2 were higher or similar to that of v686∆TRS-TRL at termination. The survival of the v686 mut2 group was significantly different from that of v686 mut1 (*p*-value = 0.039) and parental v686∆TRS-TRL (*p*-value = 0.003) (Figure 2A). However, the early mortality of the v686mut1 group was higher, and the sample size of v686mut1 was smaller than other groups (Table 4). In trial 2 with antibody-positive birds, the survival rates of v686 mut1 (29%) and v686 mut2 (53%) were higher than that of v686∆TRS-TRL (0%) at termination. The survival of both the v686 mut1 and v686 mut2 groups showed significant differences from that of v686∆TRS-TRL (*p*-value = 0.0016 with v686 mut1 and *p*-value < 0.0001 with v686 mut2) (Figure 2B).

The atrophic lesion scores for the thymus (TA) and bursa (BA) of the v686 mut1 group were similar to or lower than those of parental v686∆TRS-TRL. The TA and BA scores of the v686 mut2 group were lower than those of the v686∆TRS-TRL group (Appendix A). The frequency of tumor formation in the v686 mut1 group was lower in trial 1 but higher in trial 2. In the case of the v686 mut2 group, the frequency of tumor formation was higher than that observed with v686∆TRS-TRL in both trials. The enlargement of peripheral nerves varied between the two trials, but v686 mut1 showed a lower frequency of nerve enlargements in trial 1 when compared to the parental virus (Appendix A). The MD incidence, as measured by the atrophy of lymphoid organs, nerve lesions, or tumor formation, was similar between the v686∆TRS-TRL and v686-BAC groups. Interestingly, the MD incidence of v686 mut1 was 88% in trial 1 and 79% in trial 2, and in the case of v686 mut2, the MD incidence was lower than that of parental or other SNP-mutated virus groups (86% in trial 1 and 67% in trial 2) (Table 4). Overall, the pathogenicity of v686 mut1 and v686 mut2 was reduced compared to the v686-BAC or v686∆TRS-TRL.

### 3.3. Pathotype Analysis of the Recombinant MDVs

Three prototype strains were included to compare the virulence of the modified viruses. The survival of groups infected with the reference viruses and unvaccinated or vaccinated with vaccines with different efficacies was used to define MD protection levels for a best-fit pathotype analysis. vv+MDV 648A showed mortality at the earliest timepoint, and the mortality of vvMDV Md5 and vMDV JM/102W followed the expected virulence ranking of the viruses (Figure 3, Appendix A). The frequency and severity of MD lesions in the thymus and bursa, the enlargement of peripheral nerves, and tumor formation also agreed with the virulence ranking of MDV, in which vv+MDV showed the highest MD incidence and greatest severity of lesions among the reference MDVs (Appendix A). 

The survival rates of the v686 mut1 and v686 mut2 groups in unvaccinated birds were significantly different from that of the v686∆TRS-TRL group in Rep1 (*p*-value = 0.0004 with v686 mut1, *p*-value = 0.0001 with v686 mut2); however, no significant differences were seen in Rep2 (Figure 3). The groups challenged with the SNP-modified viruses in HVT-vaccinated or bivalent-vaccinated birds showed higher survival, but they were not statistically different from those of other challenge groups (Appendix A). MD lesions in lymphoid organs in the v686 mut1 and v686 mut2 groups were less frequent and less severe compared to the parental v686∆TRS-TRL group (Appendix A). The overall TA and BA scores in both trials indicated that the SNP modification resulted in a reduction in the atrophy of the thymus or bursa, and v686 mut2 had a greater reduction in MD lesions in the thymus and bursa (Appendix A). The enlargement of peripheral nerves was similar or less severe in the group infected with SNP-modified viruses than the v686∆TRS-TRL group in HVT- or bivalent-vaccinated birds (Appendix A). Tumor formation in vaccinated groups showed similar or less severity in the SNP-modified virus groups than in the v686∆TRS-TRL group in both monovalent- and bivalent-vaccinated birds (Appendix A).

The disease incidence in the two trials using commercial SPF chickens varied between experiments, and the overall MD incidence of Rep2 was lower than that of the Rep1 trial. The observed MD incidence with the reference strains was as expected based on the virulence, and the disease incidences in vaccinated groups more clearly discriminated the virulence ranking of MDV. The HVT vaccine showed protection against vMDV JM/102W, and the bivalent vaccine protected against the vvMDV Md5 challenge (Table 4). In non-vaccinated birds, v686∆TRS-TRL induced 100% MD in both trials, while lower MD incidences were observed in the vaccinated birds (Table 4). The reduction in MD incidence of v686 mut1 was apparent in bivalent-vaccinated groups, and that of v686 mut2 was more significant (Table 5). 

### 3.4. Graphical Display of the Recombinant MDV Pathotypes

The pathotypes of v686∆TRS-TRL and the SNP-modified viruses were analyzed using critical criteria to determine the virulence of MDV, including the MD incidence, mortality, and MD lesion scores. With the cumulative data of the two trials, v686∆TRS-TRL was classified as vvMDV or a marginal of the vv+MDV pathotype. However, v686 mut1 and v686 mut2 were categorized as vvMDV (Table 6), and the proportional distances from the reference strain pathotypes were calculated and displayed relative to the reference strains. v686∆TRS-TRL was determined to have a 0.38 propositional distance above the reference vvMDV Md5 strain (Figure 4). Interestingly, v686 mut1 was calculated to have a 0.15 proportional distance above the Md5 strain, which is the vvMDV pathotype. The distance of v686 mut2 was even closer to the Md5 strain (distance = 0.11) than that of v686 mut1 or v686∆TRS-TRL (Figure 4).

## 4. Discussion

A major goal of the current study was to validate the genomic differences associated with virulence by manipulating a vv+MDV genome to contain SNPs found in less virulent MDV strains based on a previous comparative genomics study [12]. Two independent SNP-modified viruses derived from MDV strain 686, v686 mut1 and v686 mut2, were constructed by incorporating nine nonsynonymous SNP changes in eight ORFs within a single construct to evaluate their combined contribution in attenuation. We found that the SNP-modified MDVs in multiple ORFs that encode glycoprotein H (UL22), tegument proteins (UL36, UL37), virus-host shutoff tegument protein (UL41), probable membrane protein (UL43), virus protein with an unknown function (R-LORF8), MDV oncogene (R-LORF7), and immediate early transcriptional regulator (ICP4) had reduced pathogenicity in ADOL inbred SPF chickens regardless the antibody status of the chicken. Moreover, based on their virulence, v686 mut1 and v686 mut2 were confirmed as lower pathotypes (vvMDV) compared to the highly virulent parental virus (vv+MDV) by comparative pathotype determination with the prototype strains using commercial SPF chickens. Thus, this proves that at least one of the modified SNPs contributes to virulence, and likely more, though this needs to be formally proven by testing each SNP individually. The result also indicates that the altered SNPs are not sufficient to account for differences between v- and vv+-pathotyped MDVs.

The molecular analysis of specific MDV genes has been used to identify virulence-related genes or to correlate the virulence of MDV with genetic markers [23,24,25,26,27,28,29,30,31,32,33,34,35,36]. Several MDV genes, including Meq, ICP4, pp38, vIL-8, and glycoproteins, were sequenced to predict the pathotypes of MDV isolates, and the mutations or SNPs in the Meq gene showed the highest correlation with the virulence of MDV. In a phylogenomic analysis of the MDV isolates from Eurasia and North America, a marginal correlation of mutations associated with the evolution of virulent MDVs was reported [36]. This weak correlation of mutations was detected in several ORFs (Meq, ICP4, and ICP27) from both Eurasia and North American lineages. Although efforts have been made to correlate genetic markers with MDV virulence, there are no concrete genetic markers to predict the virulence of MDV. In addition, MDV isolates whose virulence was predicted by genetic analysis did not have their pathotypes validated with a standard pathotyping assay. 

Pathotype determination, or pathotyping, is an important tool to evaluate the virulence of MDV field isolates, particularly to test new isolates from MD-vaccinated birds, but both the classical and modified best-fit pathotyping methods need a large number of birds and reference viruses with known pathotypes, as well as replicates of experiments, to make a comparative determination of the pathotypes of virus isolates [5,14,37]. In this study, we determined virus pathotypes with a modified assay using commercial SPF chickens; however, commercial SPF chickens have diverse genetic backgrounds, unlike the inbred chickens used in the traditional pathotype assay, and replicate experiments with prototype viruses of different pathotypes are required to minimize the effects of variability between experiments [34]. The SNP modifications in ICP4, UL22, R-LORF8, UL36, UL37, UL41, and UL43 within v686 mut1 and v686 mut2 resulted in identical amino acids found in the vaccine strain CVI988. Interestingly, SNP modifications of the Meq gene in v686 mut1 and v686 mut2 have an Arg119Cys change in the bZIP domain and a Gln158Pro mutation in Proline-rich repeats, which are also found in the CVI988 vaccine. However, v686 mut1 and v686 mut2 did not significantly abrogate the pathogenicity in ADOL chickens, even though all eight ORFs were modified in a single virus genome. This result indicates that the modification of two loci (119Cys and 158Pro) in the Meq gene has a minor effect on the MDV virulence, and the SNP-modified 686 viruses, v686 mut1 and v686 mut2, retained the 71Ala and 77Lys motifs found in the highly virulent MDV genome. The Meq protein, encoded by R-LORF7, is the most well-studied MDV protein, and its role in tumor formation and the virulence of MDV has been extensively reported [25,38,39,40,41,42]. Recently, Conradie et al. observed the complete abrogation of MDV virulence by replacing the Meq gene of a very virulent RB-1B strain with the short Meq gene of the CVI988 vaccine strain (vSmeq). However, the replacement with a longer Meq gene variant, known as vLmeq, from different CVI988 strains had the opposite effect and enhanced virus-induced pathogenicity and tumorigenicity [40]. The amino acid differences between vSmeq sequences of virulent RB-1B and attenuated CVI988 are nonsynonymous mutations at Ala71Ser, Lys77Glu, and Thr326Ile [26]. The virulence rank of both v686 mut1 and v686 mut2 was determined to be the vvMDV pathotype, and the propositional distance suggested that the two SNP-modified viruses had very similar virulence to the Md5 strain (Table 6). Although no evidence was found that each altered SNP in the eight ORFs is associated with virulence, the nonsynonymous SNPs suggested by previous findings and the additional SNP in UL43 were responsible for reducing virulence. An unintended SNP mutation resulted in a nonsynonymous change (Ala to Ser) in UL43. Besides the nine nonsynonymous SNPs, an additional fifteen SNPs in the intragenic and intergenic regions of the MDV genome were also identified as closely associated with virulence. The insufficient reduction to the vMDV pathotype by the SNP modification indicates that additional SNPs are likely to influence the virulence of MDV. 

In summary, 10 SNP modifications in the eight ORFs in the genome of vv+MDV resulted in reduced pathogenicity compared to the parental virus and also changed the virulence to the vvMDV pathotype. The reduced pathotype of the SNP-modified viruses indicated that the association of previously identified SNPs in the MDV genome with virulence was validated in in vivo trials with multiple sources of chickens. These findings will be useful in characterizing new MDV isolates to predict their virulence based on the SNP analysis of the marker genes and in preparing rationally designed attenuated live virus vaccine candidates by modifying multiple virulence-related genes. 

## Figures and Tables

**Figure 1 viruses-15-02263-f001:**
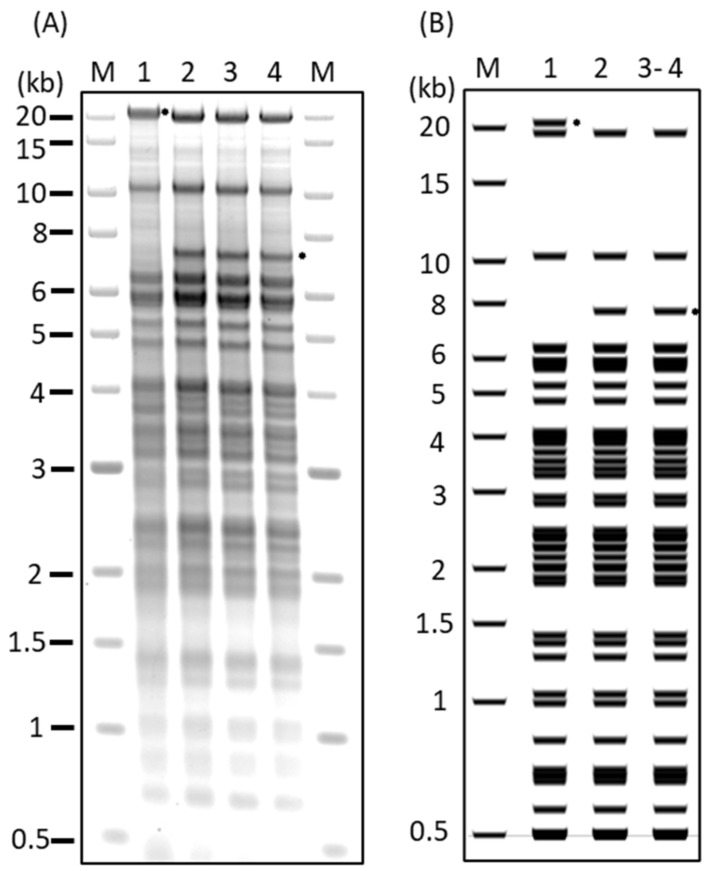
The EcoRV digestion of recombinant MDV 686-BAC clones. (**A**) Agarose gel electrophoresis image of MDV 686-BAC clones. (**B**) The expected EcoRV digestion pattern of MDV 686-BAC clones. (Lane 1) 686-BAC, (Lane 2) 686-BAC with TRS-TRL deletion (686∆TRS-TRL), (Lanes 3–4) two viruses independent of 686-BAC with nine single-nucleotide modifications (686 mut1, 686 mut2), (M) 1 kb Extend DNA ladder (NEB). The specific changes in fragmentations are on the right (*).

**Figure 2 viruses-15-02263-f002:**
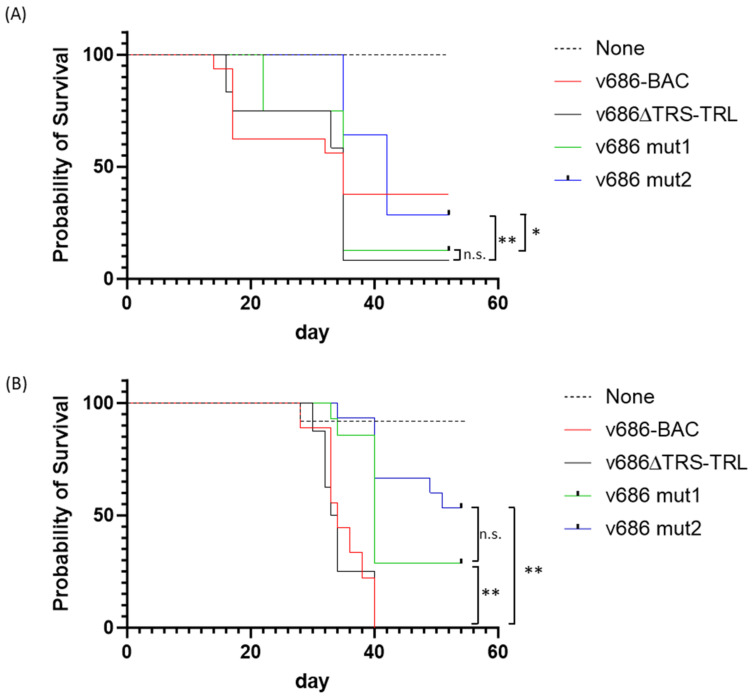
Survival rates of ADOL 15I_5_ × 7_1_ chickens infected with v686-BAC, v686∆TRS-TRL, v686 mut1, or v686 mut2. The survival rate was analyzed with the Log-rank (Mantel–Cox) test using Prism 10.0.1. (**A**) Survival percent of clinical trial 1 with maternal-antibody-negative birds; (**B**) survival percent of clinical trial 2 with maternal-antibody-positive birds. Statistical differences (*p*-value < 0.05) between v686∆TRS-TRL, v686 mut1, and v686 mut2 are indicated (n.s. = not significant, * *p*-value < 0.05, ** *p*-value < 0.01).

**Figure 3 viruses-15-02263-f003:**
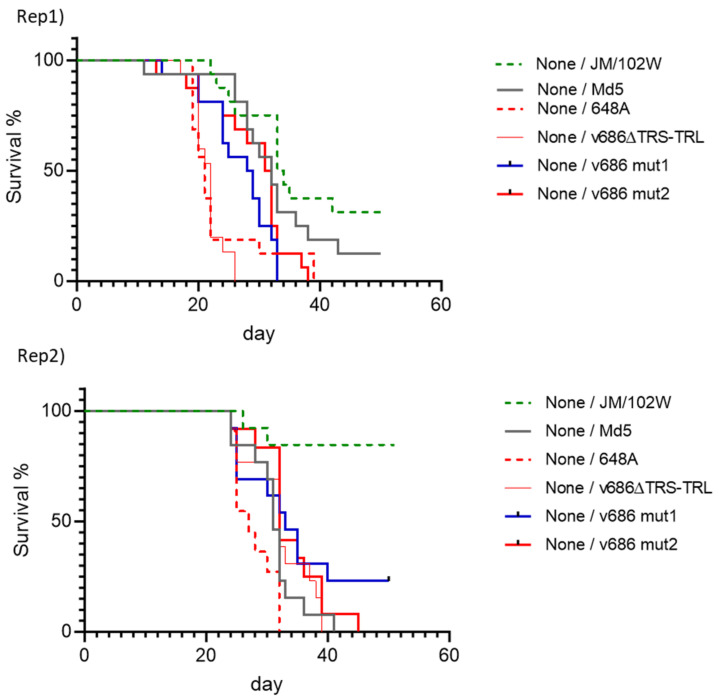
Survival curve of JM/102W, Md5, 648A, v686∆TRS-TRL, v686 mut1, or v686 mut2 challenged group in non-vaccinated birds. Survival curve of MDV JM/102W, Md5, 648A, v686∆TRS-TRL, v686 mut1, or v686 mut2 challenged group in HVT-vaccinated birds is in Appendix A and survival curve of MDV JM/102W, Md5, 648A, v686∆TRS-TRL, v686 mut1, or v686 mut2 challenged group in bivalent-vaccinated birds is in Appendix A.

**Figure 4 viruses-15-02263-f004:**
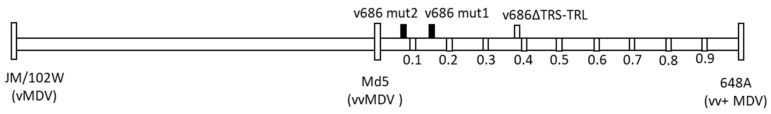
Proportional distances of the virulence of v686 mut1, v686 mut2, v686∆TRS-TRL, and the reference MDV strains with known pathotypes. The reference MDV strains are JM/102W (vMDV), Md5 (vvMDV), and 648A (vv+MDV).

**Table 1 viruses-15-02263-t001:** Modified single nucleotides from very virulent plus (vv+) MDV strain 686 genome to the nucleotides of virulent (v) MDV and its changes in amino acids.

Gene	vv+MDV	vMDV	Amino Acid Change	Protein Position
UL22	C	A	R to M	603
UL36	C	T	R to K	35
UL37	A	G	V to A	3
UL41	G	T	P to H	136
UL43	A	G	I to V	399
R-LORF8	T	C	M to V	73
R-LORF7	C	T	R to C	119
R-LORF7	A	C	Q to P	153
ICP4	A	G	S to P	181

**Table 2 viruses-15-02263-t002:** The oligonucleotides are used to modify the single nucleotides of the open reading frame (ORF) in Marek’s disease virus strain 686 genome.

ORF	Primer	Sequence (5′ to 3′) *
UL22	UL22mut.FP	ACAAGTCACAACTTCTGATGCATATATAGTCAACCTTCTC**A**TCAATGGTGTTTTAAGCTTtagggataacagggtaatcgattt
	UL22mut.RP	AGCAGAATTGGTAATCACCAAGCTTAAAACACCATTGA**T**GAGAAGGTTGACTATATATGCgccagtgttacaaccaattaacc
UL36	UL36mut.FP	TTTCGATCAAGCGAACAGAGTCCAGACGTCGAT**T**TATCTCTTAACACGTTCGCAGCTAGCtagggataacagggtaatcgattt
	UL36mut.RP	CTACATCGAACGCACCCATGCTAGCTGCGAACGTGTTAAGAGATA**A**ATCGACGTCTGGACgccagtgttacaaccaattaacc
UL37	UL37mut.FP	TCCAATAAAACTTTCAGTGGCCATATTTCATCGGTCGTTACG**G**CAGACATTATTCAGGCtagggataacagggtaatcgattt
	UL37mut.RP	ACAGAGGGTTGGCGATAGTTGCCTGAATAATGTCTG**C**CGTAACGACCGATGAAATATGGCgccagtgttacaaccaattaacc
UL41	UL41mut.FP	TCAAACCTTTCGCTCCATCTACAAACCTTCTTCCCGGGG**T**GTTTCCTCTTACACCGCGCCtagggataacagggtaatcgattt
	UL41mut.RP	GAAGTACGGATGTTGGAGAGGCGCGGTGTAAGAGGAAAC**A**CCCCGGGAAGAAGGTTTGTAgccagtgttacaaccaattaacc
UL43	UL43mut.FP	AATGCAAATAAGGGAATTAAACAATTAGCAGCTGCCTAT**G**TAGTGAAATCTATACTGGGAtagggataacagggtaatcgattt
	UL43mut.RP	GTAAACTAGTTATGATAAATCCCAGTATAGATTTCACTA**C**ATAGGCAGCTGCTAATTGTTgccagtgttacaaccaattaacc
R-LORF8	R-LORF8mut.FP	CTTCACAGGGGACATTCAAAACAAGCCCAGAGCCGTCAC**G**TGGAACACGTCTCGAGTCGAtagggataacagggtaatcgattt
	R-LORF8mut.RP	GCTTTCTTGAGGGGAGCGATCGACTCGAGACGTGTTCCA**C**GTGACGGCTCTGGGCTTGTTgccagtgttacaaccaattaacc
R-LORF7	Meqmut.FP	CTAAGGACTGAGTGCACGTCCCTGCGTGTACAGTTGGCT**T**GTCATGAGCCAGTTTGCCCTtagggataacagggtaatcgattt
	Meqmut.RP	CCGTTAGGGGTACCGCCATAGGGCAAACTGGCTCATGAC**A**AGCCAACTGTACACGCAGGGgccagtgttacaaccaattaacc
R-LORF7	Meqmut2.FP	CGCACGATCCCGTTCCTGAACCTCCCATTTGCACTCCTC**C**ACCTCCCTCACCGGATGAACtagggataacagggtaatcgattt
	Meqmut2.RP	GAGCAATGTGGAGCGTTAGGTTCATCCGGTGAGGGAGGT**G**GAGGAGTGCAAATGGGAGGTgccagtgttacaaccaattaacc
ICP4	ICP4mut.FP	CCTCACCAAAATCGGCTGCCAGGATCTCCAGTAGAGGAG**G**ACTGGATGTCCCGCCGCTTCtagggataacagggtaatcgattt
	ICP4mut.RP	GACATCGAGTCCGCTTCCGGAAGCGGCGGGACATCCAGT**C**CTCCTCTACTGGAGATCCTGgccagtgttacaaccaattaacc

* The bold nucleotide indicates that a specific nucleotide is mutated from the nucleotide of the MDV 686 strain genome. The uppercase indicates the homologous sequences for the MDV 686 strain. The lowercase indicates that the sequences were used to amplify the selection kanamycin resistance gene from pEP-KanS.

**Table 3 viruses-15-02263-t003:** Single-nucleotide polymorphisms identified with next-generation sequencing.

Gene	vv+MDV Allele	vMDV Allele	686	686∆TRS-TRL	686 mut1	686 mut2	Average Depth
UL22	C	A	C	C	A	A	126
UL36	C	T	C	C	T	T	133
UL37	A	G	A	A	G	G	131
UL41	G	T	G	G	T	T	124
UL43 *	G	T	G	G	T	T	131
UL43	A	G	A	A	G	G	133
R-LORF8	T	C	T	T	C	C	174
R-LORF7	C	T	C	C	T	T	186
R-LORF7	A	C	A	A	C	C	157
ICP4	A	G	A	A	G	G	97
LORF2 *	A	G	A	G	A	A	129

* Additional SNPs identified across all samples.

**Table 4 viruses-15-02263-t004:** MD incidence of ADOL 15I_5_ × 7_1_ SPF chickens infected with recombinant MDV or parental viruses.

Clinical Trial	Challenge Virus	Birds at Risk ^1^	MD-Positive ^2^	% MD	Differences ^3^
1	None	15	0	0	a
	v686-BAC	15	10	67	b
	v686∆TRS-TRL	12	11	92	b
	v686 mut1	8	7	88	b
	v686 mut2	14	12	86	b
2	None	12	0	0	a
	v686-BAC	9	9	100	b
	v686∆TRS-TRL	8	8	100	b
	v686 mut1	14	11	79	b
	v686 mut2	15	10	67	b

^1^ Birds that died within 7 days post-hatch with chick mortality were not included. ^2^ MD was considered when a bird had atrophy of bursa and thymus or nerve lesions or tumors. ^3^ Statistical differences between groups within each trial (*p*-value < 0.05) are indicated.

**Table 5 viruses-15-02263-t005:** Marek’s disease incidence of commercial SPF chickens infected with recombinant MDV with SNP modifications and the parental virus.

Replicate	Group Vaccine/Challenge	Birds at Risk ^1^	MD Positive ^2^	% MD	Differences ^3^
1	None/JM/102W	16	15	94	a
	None/Md5	16	14	88	a
	None/648A	16	16	100	a
	None/v686∆TRS-TRL	15	15	100	a
	None/v686 mut1	16	15	94	a
	None/v686 mut2	16	16	100	a
	HVT/JM/102W	9	2	22	a
	HVT/Md5	9	3	33	b
	HVT/648A	9	8	89	b
	HVT/v686∆TRS-TRL	10	6	60	a,b
	HVT/v686 mut1	10	6	60	a,b
	HVT/v686 mut2	9	4	44	a,b
	HVT+SB1/JM/102W	9	1	11	a
	HVT+SB1/Md5	9	3	33	a,b
	HVT+SB1/648A	9	6	67	b
	HVT+SB1/v686∆TRS-TRL	10	7	70	b
	HVT+SB1/v686 mut1	8	2	25	a,c
	HVT+SB1/v686 mut2	6	3	50	a,b
2	None/JM/102W	13	6	46	a
	None/Md5	13	13	100	b
	None/648A	11	11	100	b
	None/v686∆TRS-TRL	13	13	100	b
	None/v686 mut1	13	9	69	a,b
	None/v686 mut2	12	12	100	b
	HVT/JM/102W	16	0	0	a
	HVT/Md5	15	1	7	a
	HVT/648A	15	10	67	b
	HVT/v686∆TRS-TRL	16	2	13	a,b
	HVT/v686 mut1	16	4	25	a,b
	HVT/v686 mut2	16	3	19	a,b
	HVT+SB1/JM/102W	16	0	0	a
	HVT+SB1/Md5	15	3	20	a
	HVT+SB1/648A	16	4	25	a
	HVT+SB1/v686∆TRS-TRL	16	2	13	a
	HVT+SB1/v686 mut1	16	3	19	a
	HVT+SB1/v686 mut2	16	2	13	a

^1^ Birds that died within 7 days post-hatch with chick mortality were omitted. ^2^ MD was considered present when a bird had atrophy of the bursa and thymus or nerve lesions or tumors. ^3^ Statistical differences between groups receiving the same vaccine in the same replicate (*p*-value < 0.05) are indicated. No statistical comparison is implied between differing vaccines or across replicates.

**Table 6 viruses-15-02263-t006:** Pathotypes of the SNP-modified viruses determined by the best-fit pathotyping method.

	Non-Vaccinated			Vaccinated				Virulence by Weighted Mean	Pathotype
Virus	# of Birds	# of MD	% of MD	# of Birds Vaccinated		% of Total MD			
				HVT	HVT+SB1	HVT	HVT+SB1		
JM/102W	29	21	72	25	25	8	4	5	
Md5	29	27	93	24	24	17	25	22	
648A	27	27	100	24	25	75	40	52	
v686∆TRS-TRL	28	28	100	26	26	31	35	33	vv or vv plus
v686 mut1	29	24	83	26	24	38	21	27	vv
v686 mut2	28	28	100	25	22	28	23	24	vv

## Data Availability

Data are contained within the article and Appendix A.

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
