# Peer review of "Phenotypic Characterization of Recombinant Marek’s Disease Virus in Live Birds Validates Polymorphisms Associated with Virulence"

_viruses, 2023, doi:10.3390/v15112263_

Round 1

Reviewer 1 Report

Comments and Suggestions for Authors

In the study conducted by Kim and colleagues, it was shown if particular SNPs within specific MDV genes were associated with increased virulence. Interestingly, the authors observed a decrease in disease incidence when these SNPs were substituted with those from less virulent strains. Although these SNPs contribute to virulence, the authors rightfully discuss that there are likely other variations within the MDV genome that also play a role in this complex disease phenotype.

Major comments:

Did the authors NGS-sequenced the viruses to control for any off-target effects during mutagenesis? Or only Sanger-sequenced the genes of interest? I suggest to control for off target effects during mutagenesis – either by NGS or by using revertants. If the authors decide to sequence their mutants, it would be helpful if these sequences could be uploaded at GenBank.

The authors have to provide their statistical analyses data (asterisks and/or p values) in their figures and tables of the main manuscript as well as the SI.

It would be good if the authors provide data that shows successful in vivo infection of their chickens.

Additional minor comments and suggestions:

“Mardivirus gallidalpha 2” should be corrected (lines 17 and 35)

The role of Meq hasn’t been appropriately introduced/discussed… It would be good if the authors could add some background on this oncogene and pathotype-related signature mutations. I also suggest that the authors discuss the possible role of the meq oncogene in the evolution of higher virulence, see doi: 10.1371/journal.ppat.1009104 for example.

The authors should indicate/show data if the 686ΔTRS-TRL is restored to wt during CEF infections prior to the in vivo experiments.

Which passages of the 686 BAC and the mutants were used for the in vivo experiments? That information should be included in “2.3. Reconstitution of the 686 mut viruses”.

Should the 686ΔTRS-TRL mutant be included in Fig. 4?

Could the authors speculate why the 686 BAC seems to cause decreased (vvMDV-like) pathogenesis compared to the 686 strain?

The authors have to discuss why the survival of the v686 mut2 group was significantly different from that of the v686 mut 1.

Additional typos: “apparant” and “signficant” (line 286).

Author Response

Thank you very much for taking the time to review this manuscript. Please find the detailed responses in the attachment. 

Reviewer 2 Report

Comments and Suggestions for Authors

Marek's disease (MD) is a highly infectious lymphoproliferative disease of chickens with significant economic impact. The research on phenotypic characterization of recombinant Marek's disease virus associated with polymorphisms and virulence in live birds is fascinating. Their results support that SNPs in one or more ORFs are associated with virulence, although additional variants in the MDV genome are associated with virulence. The review suggests the minor revisions include 1) making Figure 1 clear; for example, the marker is vague in Figure 1B; 2) describing clearly the compositions of mutation types in Figure 2; 3) Missing significant tests of the survival curves in Figures 2 and 3.  

Author Response

Thank you very much for taking the time to review this manuscript. Please find the detailed responses below and the corresponding revisions highlighted in the re-submitted files. The original comments from the reviewer are in bold and our accompanying responses are in regular font. 

The review suggests the minor revisions include

  • making Figure 1 clear; for example, the marker is vague in Figure 1B;

Figure 1A has changed to a brighter gel image. The differences in fragmentation at 20.5 kb and 7.8 kb are indicated. The DNA size marker in Figure 1B has changed as suggested with a clear font.

2) describing clearly the compositions of mutation types in Figure 2;

Figure 2 caption has changed to describe each mutation type.

3) Missing significant tests of the survival curves in Figures 2 and 3. 

Statical significances were indicated the Figure 2 with asterisks. For Figure 3, we summarized the statistical differences between the same vaccine status groups in a supplementary table (Table S5).

Reviewer 3 Report

Comments and Suggestions for Authors

In this manuscript, the authors sought to determine whether a set of 9 SNPs within 8 ORFs of the Marek's Disease Virus genome confer a change in the virulence pathotype.  Indeed, using BAC-derived mutants, a transfer of all 9 SNPs to a very virulent plus strain reduced pathogenicity to a lower classification, but not entirely.  While an analysis of individual SNPs would be informative, the data supports a role for this set of SNPs in pathogenicity.  The methodology is standard and well performed, though additional replicates would benefit statistical analysis.

One additional analysis that could strengthen the paper is looking at whether any of the novel transcripts identified by global omics (https://pubmed.ncbi.nlm.nih.gov/30884829/) are especially affected by these SNPs.

Author Response

Thank you very much for taking the time to review this manuscript. Please find the detailed responses below and the corresponding revisions highlighted in the re-submitted files. The original comments from the reviewer are in bold and our accompanying responses are in regular font.

While an analysis of individual SNPs would be informative, the data supports a role for this set of SNPs in pathogenicity.  The methodology is standard and well performed, though additional replicates would benefit statistical analysis. One additional analysis that could strengthen the paper is looking at whether any of the novel transcripts identified by global omics (https://pubmed.ncbi.nlm.nih.gov/30884829/) are especially affected by these SNPs.

Thank you for this comment. We can include the transcriptome analysis to understand the potential roles of the proposed SNPs in future experiments.

Round 2

Reviewer 1 Report

Comments and Suggestions for Authors

The authors have sufficiently addressed all raised issues and improved the manuscript a lot. Congratulation on this nice work.